# Association of Self-Reported Physical Fitness during Late Pregnancy with Birth Outcomes and Oxytocin Administration during Labour—The GESTAFIT Project

**DOI:** 10.3390/ijerph18158201

**Published:** 2021-08-03

**Authors:** Laura Baena-García, Nuria Marín-Jiménez, Lidia Romero-Gallardo, Milkana Borges-Cosic, Olga Ocón-Hernández, Marta Flor-Alemany, Virginia A. Aparicio

**Affiliations:** 1Department of Nursing, Faculty of Health Sciences, University of Granada, 51001 Ceuta, Spain; lbaenagarcia@ugr.es; 2Sport and Health University Research Institute (iMUDS), 18007 Granada, Spain; lidiaromerogallardo@gmail.com (L.R.-G.); milkanaa@hotmail.com (M.B.-C.); floralemany@ugr.es (M.F.-A.); 3Department of Physical Education and Sport, Faculty of Sport Sciences, University of Granada, 18071 Granada, Spain; 4Gynaecology and Obstetrics Unit, ‘San Cecilio’ University Hospital, 18016 Granada, Spain; ooconh@ugr.es; 5Institute of Nutrition and Food Technology (INYTA), Biomedical Research Centre (CIBM), University of Granada, 18016 Granada, Spain; virginiaparicio@ugr.es; 6Department of Physiology, Faculty of Pharmacy, University of Granada, 18011 Granada, Spain

**Keywords:** umbilical cord blood, maternal health, physical fitness, flexibility, pregnancy outcomes

## Abstract

We explored (a) the associations between self-reported maternal physical fitness and birth outcomes; (b) whether self-reported maternal physical fitness (PF) is related to the administration of oxytocin to induce or stimulate labour. Pregnant women from the GESTAFIT project randomized controlled trial (n = 117) participated in this prospective longitudinal study. Maternal physical fitness was assessed through the *International Fitness Scale* at the 34th gestational week. Maternal and neonatal birth outcomes and oxytocin administration were collected from the obstetric medical records. Umbilical arterial and venous cord blood gas were analysed immediately after birth. Self-reported overall fitness, cardiorespiratory fitness, muscular strength and flexibility were not related to any maternal and neonatal birth outcomes (all *p* > 0.05). Greater speed-agility was associated with a more alkaline arterial (*p* = 0.04) and venous (*p* = 0.02) pH in the umbilical cord blood. Women who were administered oxytocin to induce or stimulate labour reported lower cardiorespiratory fitness (*p* = 0.013, Cohen’s *d* = 0.55; 95% confidence interval (CI): 0.14, 0.93) and flexibility (*p* = 0.040, Cohen´s *d* = 0.51; 95% CI: 0.09, 0.89) compared to women who were not administered oxytocin. Greater maternal physical fitness during pregnancy could be associated with better neonatal birth outcomes and lower risk of needing oxytocin administration.

## 1. Introduction

Birth and the events that occur during labour may have important implications for both, the mother and the new born’s future health. Some adverse events could be associated with natural reasons—such as placental pathologies [1] or iatrogenic causes, as a result of interventions carried out by health professionals—such as anaesthesia -related problems [2]. In this sense, it is clinically relevant to describe maternal aptitudes that may be related to an improvement in birth outcomes and a lower risk of iatrogenic injuries during labour. In relation to this, whereas the effects of exercise on birth outcomes have been broadly described [3,4], the association of maternal physical fitness (PF) with other important outcomes, such as duration of the stages of labour and values of umbilical cord blood gases, continue to be scarce and only its relationship with some maternal and foetal health markers has been previously suggested [5,6,7].

Moreover, it has been proposed in the last years to avoid interventions that are not necessary during labour [8], such as the administration of oxytocin, commonly used to increase or improve uterine dynamics [9]. It is noteworthy that the use of exogenous oxytocin during labour has been previously associated with an increased risk of uterine hyperactivity and postpartum haemorrhage, pathologies of foetal heart rate, increased morbidity and neonatal mortality and breast feeding problems, among others [10,11].

In order to study the associations of maternal PF with different health outcomes, PF can be measured in a precise way through objective tests [12,13]. However, this kind of tests requires a lot of time to carry them out, so the development of measuring tools adapted to the health professionals—who usually have less than five mins of consultation time [14] is mandatory. For this purpose, the *International FItness Scale (IFIS)* could be a validated, straightforward and handy clinical non-objective tool for assessing PF in pregnant women [15]. 

Since pregnancy and labour events have important implications for maternal and foetal health, it is of clinical interest to know which PF components may be potentially related to better birth outcomes and with a lower risk than administered oxytocin to induce or stimulate labour. Therefore, the objectives of the present study are: (i) To explore the associations between self-reported maternal PF during late pregnancy and birth outcomes; and (ii) to study whether self-reported maternal PF during late pregnancy is related to the administration of oxytocin to induce or stimulate labour.

## 2. Materials and Methods

### 2.1. Study Design and Participants

This study is part of the GESTAFIT project randomized controlled trial. Briefly, the inclusion criteria were healthy women aged 25–40 years old, with a normal pregnancy course, who signed the informed consent. Some of the exclusion criteria were high-risk obstetric pregnancies, foetal malformations and maternal malnutrition, among others (Appendix A). Detailed procedures have been previously published [16]. At the 12th gestational week (early pregnancy), a total of 384 pregnant women were informed about the project during their first visit to the gynaecologist at the ‘San Cecilio’ University Hospital and the ‘Virgen de las Nieves’ University Hospital in Granada (southern Spain). Finally, 159 women signed an informed consent which included the study aims, evaluation and procedures. 

The Clinical Research Ethics Committee of Granada, Regional Government of Andalusia, Spain approved the GESTAFIT study (code: GESTAFIT-0448-N-15, approved on 19 May 2015).

### 2.2. Procedures

The first evaluation of the study was carried out during the 16th gestational week (±2 weeks). Women completed a self-reported questionnaire on socio-demographic and clinical characteristics. Moreover, height and weight were assessed. At the 34th gestational week (±2 weeks), the second assessment of height and weight was conducted, and self-reported PF was assessed after being informed by the research team about how to properly fill it. Obstetric and gynaecological histories and birth outcomes were collected through the *Documento de Salud de la Embarazada* (from now on, *Pregnancy Health Document*, its translation into English) and digital medical records. 

### 2.3. Sociodemographic and Clinical Data

A self-reported questionnaire was used to assess sociodemographic (age, parity, number of children and educational and working statuses) and clinical (suffering or have suffered specific diseases) data. The research team was present at all times for any explanations or instructions required by the participants.

### 2.4. Anthropometry and Body Composition

Height and weight were measured using a stadiometer (Seca 22, Hamburg, Germany) and a scale (InBody R20; Biospace, Seoul, Korea), respectively. Body mass index (BMI) was calculated as: weight (kg)/height (m^2^).

### 2.5. Obstetric History

Regional Government of Andalusia provides the Pregnancy Health Document to all pregnant women. It contains obstetric and medical data that were collected for the present study. In this way, information about previous pregnancies and births and gynaecological antecedents were obtained. The gestational age was calculated by the date of the last menstruation, corrected for cycles of 28 days and subsequently corrected by ultrasound, if needed [17].

### 2.6. Birth Outcomes

All data related to the type of birth (eutocic, instrumental, or caesarean), gestational week at birth, use of epidural analgesia, length of labour, sex of the neonate, neonatal weight and Apgar test were obtained from perinatal obstetric records (partogram) collected from the Hospital. 

#### 2.6.1. Umbilical Cord Blood Gas Analysis

Between the second and the third min of the neonate’s life, a trained midwife performed a double clamping of the umbilical cord, with a minimum distance between both clamps of 10 cm for the umbilical cord blood sampling. A pre-heparinized 1 mL syringe was used for the blood extraction. Blood samples were taken from both, umbilical artery and vein. Partial pressure of carbon dioxide (PCO2), partial pressure of oxygen (PO2), oxygen saturation and pH were analysed using a blood analyser (GEM Premier 4000, Instrumentation Laboratory, Bedford, MA, USA).

#### 2.6.2. Oxytocin Administration before or during Labour

Information about the use of oxytocin during labour was collected from the partogram. In this document, midwives usually record whether oxytocin is administered or not, but the dose and administration time are not frequently collected, so these data were not assessed in the present study. Moreover, we consider that oxytocin was administered both, by induction of labour (for clinical reasons such us post-term pregnancy, premature rupture of membranes or preeclampsia, among others) and uterine stimulation during the labour, but we did not take into account the administration of this drug during placenta delivery. 

### 2.7. Physical Fitness

Physical fitness was self-reported with the International FItness Scale (IFIS), which is composed of five Likert-scale questions from 1 (very poor) to 5 (very good) asking the participants about their perceived overall PF, cardiorespiratory fitness (CRF), muscular strength (MS), speed-agility and flexibility in comparison with their counterparts. The participants rated their PF components as ‘very poor’, ‘poor’, ‘average’, ‘good’ and ‘very good’ [18]. The higher score indicates greater PF that the participant experiences. This questionnaire is available in different languages at http://profith.ugr.es/IFIS (accessed on 20 January 2021).

### 2.8. Statistical Analyses

All the variables were checked for normality of distribution before the analyses. We employed descriptive statistics [mean (standard deviation, SD)] for quantitative variables and number of cases and percentage (%) for categorical variables to describe the baseline characteristics of the study sample (Table 1). The association of self-reported PF with birth outcomes was assessed with Pearson´s partial correlations after adjusting for maternal age, parity, maternal BMI and exercise intervention (Table 2). The associations of PF levels with duration of first and second labour stages, as well as neonatal birth outcomes were additionally adjusted for epidural analgesia, except for the variable ‘birthweight’ which was only further adjusted for gestational age. Since some variables (adherence to the Mediterranean diet, gestational age at birth, birthweight, onset of labour and instrumental birth) previously showed slight relationship with the study variables, we performed secondary analyses to assess their role as potential confounders. 

An analysis of the covariance (ANCOVA) was employed to explore the differences in self-reported PF components between women who were administered versus women who were not administered oxytocin during labour (Table 3), and it was adjusted for maternal age, parity, maternal BMI, epidural analgesia and birthplace (public or private hospital). Elective caesarean sections (elected because of feet or buttocks coming first, n = 6) were excluded from the present analyses. Finally, like in the GESTAFIT project [16], a concurrent physical exercise program was performed, we have also adjusted all analyses for the exercise intervention (control or intervention). Furthermore, standardised effect size statistics were estimated in all the comparisons through Cohen’s *d* and its exact confidence interval (CI). The effect size was interpreted as small (∼0.2), medium (∼0.5) or large (∼0.8 or greater) [19]. The statistical analyses were conducted with the Statistical Package for Social Sciences (IBM SPSS Statistics for Windows, Version 20.0. Armonk, NY, USA: IBM Corp). The statistical significance was set at *p* < 0.05.

## 3. Results

From the 159 women who signed the informed consent and met the inclusion criteria, 158 had complete sociodemographic data. The final sample size was composed of 117 Caucasian pregnant women (age 33.1 ± 4.5 years old, BMI 24.6 ± 4.1 kg/m^2^) who presented valid data for the present analyses (Figure 1).

Sociodemographic and clinical data, and self-reported PF components of the study sample are shown in Table 1. Most of the participants lived with their partners (98.3%) and more than a half of them had university degrees (63.2%). Broadly, participants rated their PF levels as ‘*average*’, except for CRF, which was classified as ‘*poor*’, compared to their peers. Regarding birth outcomes, 77.3% had vaginal births and nearly a quarter had caesarean sections. Around 58% of participants were nulliparous and 27.8% were administered oxytocin before or during the labour. They gave birth around the 40th gestational week and neonates had a birthweight average of 3338 ± 428 g.

Pearson´s partial correlations of self-reported PF levels at the 34th gestational week with pregnancy and neonatal birth outcomes are shown in Table 2. After adjusting for potential confounders, speed-agility showed a significant positive association with arterial (*p* = 0.04) and venous cord blood pH (*p* = 0.02). Greater overall PF and MS showed some evidence of borderline statistical significance with higher cord blood venous pH (both, *p* < 0.100). Nevertheless, flexibility was not associated with any birth outcomes (all, *p* > 0.05). Finally, maternal outcomes, birthweight, Apgar test and partial pressure of CO_2_, O_2_ and oxygen saturation in both umbilical blood vessels were not associated with any self-reported PF component (all, *p* > 0.05).

Differences on self-reported PF levels at the 34th gestational week of the study participants by oxytocin administration before or during labour are shown in Table 3. Women who were not administered oxytocin showed greater CRF compared to women who needed oxytocin (of 2.73 ± 0.7 versus 2.32 ± 0.8, respectively, *p* = 0.044 for the unadjusted model and *p* = 0.013 for the adjusted model, Cohen´s *d* = 0.55; 95% CI: 0.14, 0.93). Greater flexibility was shown in women who were not given oxytocin compared to women who were given oxytocin (of 3.27 ± 1.1 versus 2.75 ± 0.9, respectively, *p* = 0.013 for the unadjusted model and *p* = 0.040 for the adjusted model, Cohen´s *d* = 0.51; 95% CI: 0.09, 0.89). No significant differences were found in self-reported overall fitness, MS and speed-agility between women who were not administered and those who were administered oxytocin during the labour (all, *p* > 0.05).

## 4. Discussion

This study explored the association of maternal self-reported PF levels in late pregnancy with birth outcomes, and the use of oxytocin during the labour. One of the major findings of the present study is that those women who required oxytocin during labour showed lower self-reported CRF and flexibility in late pregnancy. Moreover, greater maternal speed-agility was associated with a more alkaline pH in both, umbilical artery and vein. 

In this study, no association was found between maternal gestational week at birth, and duration of first and second stages of labour with PF components. As far as we know, there are no previous studies analysing the influence of self-reported maternal PF on birth outcomes, so the reliable comparison of the present findings is extremely difficult. Nevertheless, our results do not concur with those described by Kardel et al. [20], who observed a relationship of CRF (objectively measured through maximum oxygen consumption) at the 35–37th gestational weeks, with shorter labours in 40 nulliparous women. It should be taken into account that the mean CRF of our sample was reported as ‘poor’, and it is possible that the shortening of labour was related to slightly higher values of this PF dimension. On the other hand, two studies have previously found a positive association of maternal MS with birthweight in two samples of 95 and 65 healthy pregnant women respectively [5,21], but in both cases the importance of the second trimester of pregnancy is highlighted for this association to be positive. It is possible that this finding was not found in late pregnancy, since in the third trimester maternal MS tends to decrease [5].

Notwithstanding, we found that greater self-reported speed-agility was related to more alkaline pH in both umbilical vessels. Preventing foetal acidosis is essential for the future health of the new born, since arterial pH under 7.20 has been related to an increased number of neonatal pathologies and deaths [22]. During labour, many women experience fatigue, which is associated with both, physical and mental discomfort [23]. For this reason, we hypothesise that women who feel themselves more agile might be more likely to have an active labour with greater changes of body position, that leads to an improvement of birth outcomes [24,25]. These results should be interpreted with caution and more studies are needed to confirm these associations.

After adjusting for substantial potential confounders, we found that women who were administered oxytocin during labour had lower CRF and flexibility than women who did not need this drug. CRF is related to the ability of the circulatory and respiratory systems to supply oxygen to the tissues and to eliminate fatigue products [26] and it has been positively associated with endogenous oxytocin levels in pregnant women [27]. Moreover, labour is a strenuous process that often has a long duration and usually produces a huge physiological fatigue [23]. In this sense, it has been described that fatigue during labour is related to alterations in uterine dynamics [28], which is one of the indications for the administration of synthetic oxytocin [29]. Although more studies are needed to confirm this, it is possible that women with greater CRF experienced less fatigue during labour and, therefore, showed a better uterine dynamic, so that they did not need exogenous oxytocin administration. 

Regarding flexibility, no previous studies have associated this PF dimension with the use of oxytocin. A possible explanation to our findings could be that women with greater self-reported flexibility during pregnancy could also have higher levels of relaxin, which is naturally increased during pregnancy [30]. Thus, as this hormone has vasodilator effects when acting on its receptors in the uterine artery [31], it is possible that the increase in uterine vascularization improves its dynamics during labour. Finally, it seems that high levels of relaxin could also have an important implication in the appearance of uterine contractions during pregnancy [32,33], although the mechanisms are not fully understood yet [34]. 

According to the data of the Spanish Ministry of Health, Social Services and Equality, the prevalence of the use of exogenous oxytocin during spontaneous labour in Spanish public hospitals is 53%, much higher than the recommended standard of 5–10% [35]. Among the women who presented valid data for the analysis of differences in PF by oxytocin administration, almost one in four had been provided with this hormone during labour, so these figures are still higher than recommended. Synthetic oxytocin is widely used as a treatment for dystocia of uterine dynamics [29] and as a method to induce the labour [36]. However, its use has been related to an increased risk of uterine hyperactivity, abnormalities in the foetal heart rate and postpartum haemorrhage [10]. In addition, other studies have associated the use of oxytocin during birth with sucking problems of the neonate and early cessation of breast feeding [37], among other neonatal problems [38]. 

Altogether, the findings of the present study are clinically relevant. The *IFIS* could be employed as an easy and quick clinical tool by midwives and other health professionals to assess women who should receive health education, in order to improve their PF and avoid, in this way, the potential need to be administered oxytocin before or during the future labour. Nevertheless, more studies are needed to confirm the findings of our study.

Some limitations of the present study should be highlighted. First, the associations found cannot be explained by a direct causality, since the results are derived from a cross-sectional study design. Therefore, further research is warranted in order to address the causality of our findings. Second, although subjective measures tend to overestimate some dimensions, the IFIS has been validated in the pregnant population [15].

Key strength of our methodology is that the sample size employed was relatively large compared to other studies performed on pregnant women. Further strengths include the gas measurement of the umbilical artery blood is considered a gold standard, and the extraction of paired samples (in both umbilical vessels) allowed a more accurate interpretation of the results.

## 5. Conclusions

Overall, in late pregnancy, we found that greater self-reported speed-agility was related to more alkaline pH in arterial and venous umbilical cord blood gas. Moreover, women who were administered oxytocin during labour reported lower self-reported CRF and flexibility during late pregnancy than women who were not given this drug, which indicates that greater levels in both PF components could be related to a decrease of the risk of having an induced or stimulated labour with oxytocin.

Assessing PF during pregnancy and encouraging women to improve it by performing physical activity and exercise could have important implications for the well-being and health of women and their children. In order to decrease the already high rates of oxytocin administration before or during labour in Spain, interventions focused on improving physiological births should be promoted.

## Figures and Tables

**Figure 1 ijerph-18-08201-f001:**
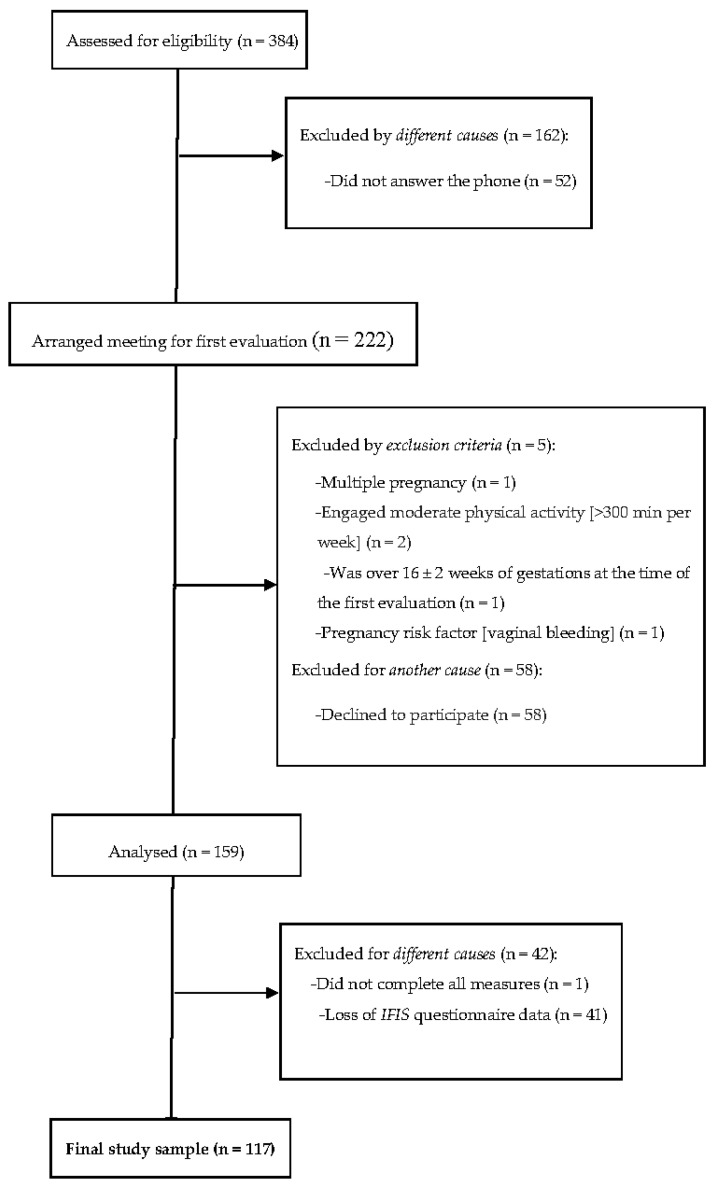
Flow diagram of study participants.

**Table 1 ijerph-18-08201-t001:** Sociodemographic and clinical characteristic of the study sample (n = 117).

Maternal Outcomes	n	Mean (SD)
Age, years	117	33.1 (4.5)
Body mass index at 16th gestational week, Kg/m^2^	117	24.6 (4.1)
		n (%)
Living with a partner		115 (98.3)
Educational status	117	
Primary or high-school		24 (20.6)
Specialized training		19 (16.2)
University degree		74 (63.2)
Working status	117	
Homework/unemployed		34 (29.1)
Partial-time employed/student		34 (29.1)
Full-time employed		49 (41.8)
Self-reported Physical Fitness (0–5)		
34th gestational week	117	
Overall physical fitness		3.3 (0.7)
Cardiorespiratory fitness		2.6 (0.8)
Muscular strength		3.3 (0.7)
Speed-agility		3.0 (0.7)
Flexibility		3.1 (1.0)
Type of birth	110	
Miscarriages	117	0 (0)
Spontaneous Instrumental vacuum/forceps		65 (59.1) 20 (18.2)
Caesarean section		25 (22.7)
Oxytocin administered during labor, n %		30 (27.8)
Birth place	115	
Public Hospital		108 (93.9)
Private Hospital		6 (5.2)
Home		1 (0.9)
Parity	117	
Nulliparous		68 (58.1)
Multiparous		49 (41.9)
Neonatal outcomes	110	
Sex (female, n (%))		55 (50)
Gestational age at birth, wk		39.5 (1.2)
Birthweight, g		3338.5 (428.7)
Apgar Test 1 min		8.6 (1.1)
Apgar Test 5 min		9.6 (0.7)
Umbilical Cord blood Gas		
Arterial pH	80	7.2 (0.07)
Arterial Partial Pressure CO_2_, mmHg	75	51.6 (10.7)
Arterial Partial Pressure O_2_, mmHg	72	19.5 (9.4)
Arterial O_2_ saturation, %	69	34.6 (22.8)
Venous pH	93	7.3 (0.1)
Venous Partial Pressure CO_2_, mmHg	89	39.7 (7.7)
Venous Partial Pressure O_2_, mmHg	80	26.2 (8.5)
Venous O_2_ saturation, %	77	54.7 (18.0)

Values shown as mean (SD, standard deviation) unless otherwise indicated; CO_2,_ carbon dioxide; O_2_, oxygen.

**Table 2 ijerph-18-08201-t002:** Partial correlations of self-reported physical fitness component levels at 34th gestational week with maternal and neonatal birth outcomes.

	Overall Fitness	Cardiorespiratory Fitness	Muscular Strength	Speed-Agility	Flexibility
Maternal outcomes					
Week of gestation (at birth) (n = 101)	0.018	0.083	0.087	0.070	0.014
Duration of first stage of labour ^a^ (n = 63)	−0.014	−0.063	0.013	0.061	0.085
Duration of second stage of labour ^a^ (n = 69)	−0.021	−0.069	−0.040	0.103	0.152
Neonatal outcomes					
Birthweight (n = 101)	−0.009	−0.112	0.053	0.070	0.003
Apgar Test 1 min ^a^ (n = 95)	0.101	0.075	−0.072	0.034	−0.048
Apgar Test 5 min ^a^ (n = 95)	−0.081	−0.042	0.002	−0.107	−0.177
Cord blood arterial pH ^a^ (n = 69)	0.109	0.056	0.154	0.232 *	−0.017
Cord blood arterial partial pressure of CO_2_ ^a^ (n = 65)	−0.044	0.005	−0.094	−0.165	0.049
Cord blood arterial partial pressure of O_2_ ^a^ (n = 62)	−0.006	−0.156	0.042	−0.020	−0.102
Cord blood arterial oxygen saturation ^a^ (n = 59)	−0.051	−0.110	0.045	−0.023	−0.130
Cord blood venous pH ^a^ (n = 81)	0.207	0.091	0.214	0.239 *	−0.028
Cord blood venous partial pressure of CO_2_ ^a^ (n = 78)	−0.074	0.024	−0.183	−0.130	0.040
Cord blood venous partial pressure of O_2_ ^a^ (n = 69)	0.094	−0.062	0.130	0.133	0.034
Cord blood venous O_2_ saturation ^a^ (n = 66)	0.127	0.011	0.108	0.033	0.109

O_2_, Oxygen; CO_2,_ carbon dioxide. Model adjusted for maternal age, parity, maternal body mass index at the 16th gestational week, and the exercise intervention. Maternal birth outcomes were additionally adjusted for birthweight. ^a^ Model additionally adjusted for epidural analgesia. Birthweight was further adjusted for gestational age. * *p* < 0.05.

**Table 3 ijerph-18-08201-t003:** Differences in the self-reported physical fitness components of the pregnant women at the 34th gestational week by oxytocin administration before or during labour.

	Oxytocin Was Not Administered (n = 78)	Oxytocin Was Administered (n = 30)	*p*	*p* *	Effect Size *d*-Cohen (95% CI)
Overall fitness	3.38 (0.8)	3.25 (0.7)	0.067	0.348	0.16 (−0.22, 0.55)
Cardiorespiratory fitness	2.73 (0.7)	2.32 (0.8)	0.044	0.013	0.55 (0.14, 0.93)
Muscular strength	3.27 (0.7)	3.29 (0.7)	0.966	0.982	−0.014 (−0.37, 0.35)
Speed-agility	2.95 (0.7)	2.96 (0.6)	0.967	0.728	0.0 (−0.51, 0.37)
Flexibility	3.27 (1.1)	2.75 (0.9)	0.013	0.040	0.51 (0.09, 0.89)

CI, Confidence interval. * Model adjusted for maternal age, parity, maternal body mass index at the 16th gestational week, exercise intervention, epidural analgesia and birthplace. Values shown as mean (standard deviation).

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
