# Peer review of "Association of Self-Reported Physical Fitness during Late Pregnancy with Birth Outcomes and Oxytocin Administration during Labour—The GESTAFIT Project"

_ijerph, 2021, doi:10.3390/ijerph18158201_

Round 1
Reviewer 1 Report
This is a very interesting study. The work could be improved by more rigorous presentations of the results, as well as some explanations and clarification.
-The study only enrolled women who were pregnant up to 34th gestational week. It is interesting to know whether there is any difference between early and late pregnancy in terms of the association between physical fitness and maternal/neonatal outcomes, as well as the association between physical fitness and oxytocin administration.
-In many cases, physical activity is a longitudinal variable which varies from early to late pregnancy. Physical fitness level in late pregnancy could be impacted by physical fitness level in early or mid- pregnancy. Thus, the association between physical fitness and birth outcomes observed during late pregnancy could be confounded by physical fitness level prior to or during early/mid pregnancy. Would it be possible to adjust physical fitness history in the model?
-Based on the flowchart, it is unclear whether women who were assessed for eligibility at the beginning of the study had any miscarriage by 34th gestational week. I wonder what the miscarriage rate was in this study compared to the general population and who likely the results could be sensitive to survival bias, i.e. would it be possible that pregnancies by 34th gestational week represent a relatively healthy women with better physical fitness status and lower risks for birth outcomes.
-Based on the flowchart, at least 50% of women declined to participate in the study. Given the response rate, how likely that these non-participation women were missing at random?
-Does the study population enrolled from the two university hospitals represent the total pregnant population? For example, would it be possible that women who visited these two hospitals were at higher risks of birth outcomes? How the results can be generalized to the total population?
-What the Physical fitness level compared to the general population?
Author Response
July 26th, 2021
Dear reviewer,
Please, find enclosed the revised version of our manuscript entitled, “Association of self-reported physical fitness during late pregnancy with birth outcomes and oxytocin administration during labor. The GESTAFIT project." by Dr. Baena-García et al., to be considered for publication in International Journal of Environmental Research and Public Health. We would like to gratefully thank the Reviewer for his/her thoughtful and constructive comments, which have undoubtedly improved the quality of our manuscript. We have carefully considered all of the suggestions, and have integrated them into the revised manuscript. Changes to the original manuscript have been incorporated by using yellow background. We believe our manuscript is now stronger as a result of these modifications. An itemized point-by-point response to the reviewers’ comments is presented below.
"Please see the attachment"

Reviewer 2 Report
The intervention in physical activity (PA) during pregnancy and its impact on both gestational and neonatal health is interesting. Few studies show clear evidence on what type of PA is required to improve health conditions during pregnancy. However, there are some points throughout the text that are not clear and make it difficult to read.
At first it seems that the article focuses on looking at PA progression over time (week 16 to 34) but then, the oxytocin administration variable determines if adequate PA is associated with gestational outcomes. If the dependent variable is oxytocin administration, Table 1 should be segmented according to this variable and determine what other baseline variables are affecting. Thus, the models (Table 3) can be controlled for those variables. In this regard, it is not clear to me which variables were dependent and which were independent.
Figure 1 should be included in the material and methods. In addition, the inclusion and exclusion criteria should also be included in the main text. What is the PARmed-X questionnaire? On the other hand, there are certain exclusion criteria that should be commented on, for example, why were multiple pregnancies excluded? Was the growth restriction of the uterus or of the fetus?
On the other hand, it is not clear in the procedure whether any type of PA intervention was applied, how was carried out and which group was selected. Also, some variables are confusing, what is the difference between the gestational age at birth (line 116) and the one calculated in the obstetric history (line 111)? Were the anthropometric measurements used to fit the models? Were these measurements collected in two times?
I would like to comment that, although the statistical section is very well described and gives details, I have doubts about the adjustment of the partial correlations in table 2 (line 234). The partial correlation is not exactly adjusted rather than determine what variance two variables explain on a third variable. Why was it decided to do partial correlations instead of Spearman or Pearson correlations?
Minor comments:
- Review the guidelines of the journal, in the names, mayor-title is no required.
- In the abstract "PF" is not defined before. In addition, report the exact p-value in the abstract and long the text.
- Keyword: "neonate" is not appropiate for this article.
Author Response

(The authors gave the same response as above.)

Reviewer 3 Report
Only minor changes is necessary for this manuscript:
1) Please indicate exact p-values in Abstract and Tables.
2) Is it necessary to apply multiple comparison corrections? Please response in the response letter.
3) Please indicate the sample size used by Kardel et al, as well as for other studies used to compare these results.
Author Response

(The authors gave the same response as above.)

Round 2
Reviewer 2 Report
I would to thank the authors for this improved version of the manuscript.